# Goblet Cell Carcinoma of the Appendix with Synchronous Adenocarcinoma of the Cecum: Distinct or Related Entities?

**DOI:** 10.3390/diseases10040077

**Published:** 2022-10-03

**Authors:** Leonardo Vincenti, Valeria Andriola, Gerardo Cazzato, Anna Colagrande, Felicia Fiore

**Affiliations:** 1Section of General Surgery, Policlinico Universitario di Bari, 70124 Bari, Italy; 2Section of Molecular Pathology, Department of Emergency and Organ Transplantation (DETO), University of Bari “Aldo Moro”, 70124 Bari, Italy

**Keywords:** goblet cell carcinoma, second primary malignancies, neuroendocrine tumors

## Abstract

Goblet cell carcinoma (GCC) is a rare primary tumor of the appendix characterized by both epithelial and neuroendocrine components containing goblet cells. While in the past, the GCC has been associated with neuroendocrine tumors, recent studies consider that GCC is closer to adenocarcinoma than a neuroendocrine component. The association between gastro-intestinal (GI) carcinoids and second primary malignancies (SPMs) is widely described in the literature, but there is no reported case of GCC and synchronous adjacent adenocarcinoma of the colon. We describe the first case in the literature, to our knowledge, of synchronous colorectal adenocarcinoma of the cecum and GCC of the appendix that are incidentally discovered in the resected primary cancer specimen. The association between the two neoplasms seems to be not causal and maybe the “paracrine-effect theory” may explain the development of a second tumor close to the primary.

## 1. Introduction

Goblet cell carcinoma (GCC) is a rare tumor, almost exclusively seen in the appendix, characterized by both epithelial (adenocarcinoma) and neuroendocrine components containing goblet cells, and it produces neuroendocrine markers and mucin. GCC is often asymptomatic and sometimes it is incidentally discovered during the routine management of acute appendicitis. Clinical presentation is variable and can either be an acute right fossa iliac syndrome or a more chronic abdominal pain syndrome with a palpable pelvic mass. It is found in 0.3% to 0.9% of appendectomies, accounting for 35–58% [1,2] of overall appendiceal lesions and ~14% of all malignant neoplasms of the appendix [3]. The association between gastro-intestinal (GI) carcinoids and second primary malignancies (SPMs) is widely described in the literature, but there is no reported case of GCC and synchronous adjacent adenocarcinoma of the colon. While in the past, GCC has been associated with neuroendocrine tumors, recent studies consider GCC closer to adenocarcinoma than a neuroendocrine component, especially from a biological and prognostic point of view.

## 2. Case Presentation

A 72-year-old female was transferred to our Unit of General Surgery from the department of internal medicine, in which she was hospitalized for bilateral pleural effusion and impaired nutritional status. Her past medical history reported ischemic heart disease, kidney failure and previous chronic myeloid leukemia, and is currently in clinical remission. During hospitalization, she underwent colonoscopy and gastroscopy. The physical examination was normal. Laboratory tests showed Hb: 11 g/dL, while CEA had a normal value. Cromogranin-A and neuro-specific enolase (NSE) were not evaluated. A colonoscopy revealed a polypoid villous lesion in the cecum, measuring 3.5 cm in diameter, but an endoscopic resection was not feasible due to its characteristics. The biopsy found adenocarcinoma, but no metastatic lesions were found on the computed tomography scan (TC) of the chest and abdomen. Thus, the patient was admitted to our unit and underwent an elective right laparoscopic colectomy with intracorporeal side-to-side anastomosis. The post-operative recovery was uneventful. The patient was discharged on post-operative day 5. A sum of clinical features is present in Table 1.

The histopathological report unexpectedly found two different primary tumors at a distance of 2.5 cm from each other: a well-differentiated adenocarcinoma of cecum pT1N0M0 (degenerated adenomatous polyp) (Figure 1) and a GCC of the appendix pT3N0M0, stage II (25–50% pattern high grade), according to the WHO 2019 (Figure 2, Figure 3 and Figure 4). Tumor cells showed a positive reaction for neuroendocrine markers such as synaptophysin (Syn), Chromogranin-A (CgA) (Figure 5), and for markers, CDX2 (Figure 6), CK19 and CK20. Perineural invasion was reported. No metastatic lymph nodes were reported on the 16 retrieved, and the surgical resection margins were negative for malignant cells.

## 3. Discussion

Most of the appendiceal tumors are carcinoid, followed by adenocarcinoma and mucinous adenocarcinoma, while GCC is a rare primary tumor of the appendix, and rarely occurs in other sites. Appendicular GCCs are more aggressive than carcinoid appendiceal tumors but less aggressive than adenocarcinomas, and they often present with serosal and mesenteric involvement [4]. Pathogenesis remains unclear: it seems that GCC derives from the pluripotent intestinal cryptoglandular stem cells which have the potential to differentiate both as mucinous and neuroendocrine types (unitary intestinal stem cell theory) [5]. Recent studies, based on immunohistochemical and molecular findings, have shown that the expression in these tumors of CEA, CDX2, CK7 and CK20 is similar to that of colonic adenocarcinomas [6]. The most common site of GCC is the appendix, with an incidence of 1.2 cases out of 1,000,000 people for years. Even though the behavior of GCC more closely resembles colorectal adenocarcinomas than neuroendocrine tumors, goblet cell neoplasms are characterized by a mutational profile that is distinct from both tumors.

In the WHO Classification of Tumours 5th edition “Digestive System Tumours”, GCC is classified as a distinct entity (so-called appendiceal goblet cell adenocarcinoma) [7]. It is an amphicrine tumor composed of goblet-like mucinous cells, with variable numbers of endocrine and Paneth-like cells [8]. Histopathologically, the tumor involves the wall of the appendix circumferentially, usually without eliciting a stromal reaction, and it is composed of small, rounded nests of signet ring-like cells resembling normal intestinal goblet cells, except for the nuclear compression. Extracellular mucin is often present and sometimes abundant. GCC is graded using a three-level system, based on the combination of low-grade and high-grade patterns of differentiation.

The AJCC guidelines consider GCC as epithelioid carcinoma and use the TNM classification of appendiceal adenocarcinoma [9]. Stage is the most important prognostic factor: the prognosis is good for the stages I-II (disease-free survival at 5 years of 100% and 76%, respectively) but poor in III or IV stages (22% and 14%, respectively) [1]. The post-operative treatment of patients with GCC is challenging due to their rarity and a poor understanding of their biological behavior. This involves the lack of evidence-based therapeutic algorithms for optimal treatment. The role of diagnostic biomarkers is limited because GCCs are diagnosed incidentally subsequent to an appendicectomy. Peritoneal carcinomatosis is the most common cause of death. Blood markers such as chromogranin A and B did not show usefulness for monitoring recurrence or disease status, while CEA, Ca 125 and CA19.9 are effective during follow-up. Lymph node metastasis, positive resection margins, extra-appendiceal spread, increased mitoses (>2 per 10 high-power fields) and a large (>50%) adenocarcinoma component are negative prognostic factors. If the carcinomatous component of the tumor exceeds 50% of the total volume, the prognosis is significantly worse. This indicates that the aggressive behavior of the GCC is related to the adenocarcinoma component rather than the neuroendocrine. Thus, a careful histologic assessment is essential to identify and quantify the percentage and grade of an adenocarcinomatous component within the tumor. Cancer survivors are prone to developing additional subsequent cancers [10]. From 20 to 63% of patients with GCC present with metastatic disease [11]. Common metastatic sites include the peritoneum and omentum. The ovary is a common metastatic site in women [11]. Lymph node metastases have been detected in 22–38% of cases [12]. Metastasis to solid organs, such as the liver, bone, and brain, is uncommon [13]. According to the study conducted by Yozu et al., the 10-year survival rates for low grade, intermediate grade, and high-grade tumors were 78%, 33%, and 4%, respectively [9].

The association of GI carcinoid and colorectal cancer is already described in the literature, but the association between GCC and adenocarcinoma has never been described before. Several studies showed that the diagnosis of colorectal carcinoid is often associated with the development of second primary malignancies (SPMs), especially as a synchronous lesion. The incidence of all SPMs range from 12% to 46% [14,15]. A recent population-based analysis demonstrated a significant excess risk of subsequent cancers localized in the small intestine followed by the colon/rectum in patients affected by appendiceal cancer [16]. Based on histological subtypes of appendiceal cancer, the greatest risk of additional cancer was observed with goblet cell carcinoid (SIR, 1.55; 95% CI, 1.23–1.92) and adenocarcinoma (SIR, 1.42; 95% CI, 1.18–1.69) [16].

The most accredited pathogenetic hypothesis at the origin of the association between carcinoids and SPMs is the theory of the paracrine effect [14]: carcinoid cells produce neuroendocrine peptides that act as growth factors on neighboring cells, starting a carcinogenesis pathway. Furthermore, intestinal cells can overexpress receptors for these regulatory peptides, promoting tumor growth in predisposed organs. The colon is among them for the same embryonic visceral origin.

A carcinoid component also produces non-neuroendocrine peptides that may play a role in carcinogenesis. The transforming growth factors are elaborated in the gut and bind to specific high-affinity receptors on fibroblasts or endothelial cells. Their principal role is to regulate cell growth and differentiation, including angiogenesis and blood vessel ingrowth in tumors [17]. PDGF, EGF, TGF, insulin-like growth factors and FGF recently have been demonstrated in gastrointestinal carcinoids, and these growth factors may play a central role in the genesis of SPMs in patients with carcinoid tumors.

However, in our case, the neuroendocrine component represents only 30% of the GCC, while the epithelial component is prevalent (70%). The onset of SPMs might be explained by the presence of neuropeptides in case of carcinoid tumors; however, in this case, considering the prevalence of the epithelial component, it is not possible to determine if the two cancers could be considered as distinct or related entities.

## 4. Conclusions

In conclusion, it has been widely demonstrated that carcinoid tumors are associated with SPMs in most cases located in the GI tract. We describe the first case in the literature of synchronous colorectal adenocarcinoma of the cecum and GCC of the appendix that are incidentally discovered in the resected primary cancer specimen. Although GCC has a small neuroendocrine component and has a behavior more similar to adenocarcinoma in terms of staging and prognosis, nevertheless, it could promote the onset of an SPM, exactly as it happens for carcinoid tumors.

The association between the two neoplasms seems to be not causal and maybe the “paracrine-effect theory” may explain the development of a second tumor close to the primary (2.5 cm), as well as the slow evolution of the normal colonic mucosa to tubular villous polyps and finally to adenocarcinoma. However, the identification of a GCC requires the plan of a systematic follow-up to identify early SPM onset.

## Figures and Tables

**Figure 1 diseases-10-00077-f001:**
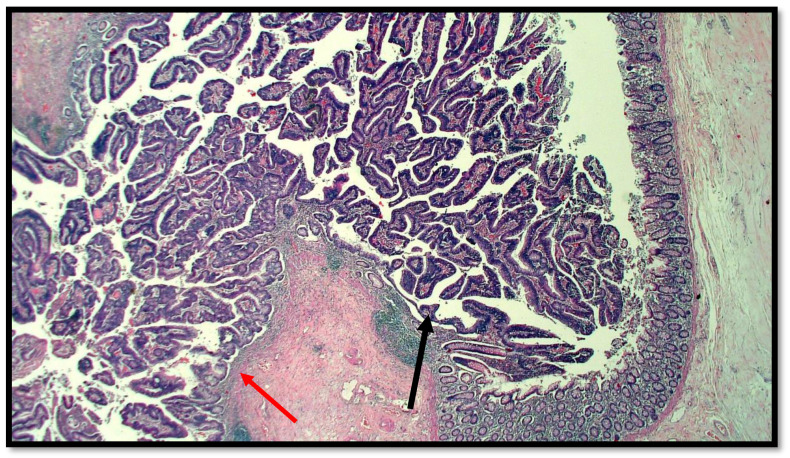
Histological preparation of intestinal-type adenocarcinoma (black arrow) arising on villous adenoma with high grade dysplasia (red arrow) (hematoxylin–eosin, original magnification: 10×).

**Figure 2 diseases-10-00077-f002:**
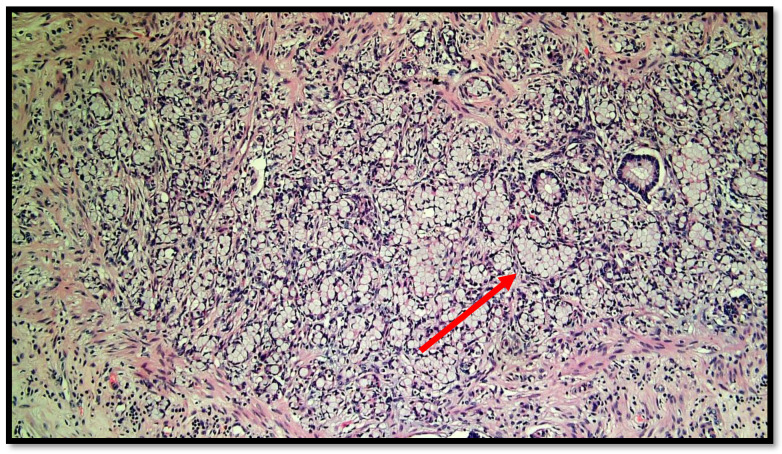
Photomicrograph showing goblet cell carcinoid of the appendix. Cytological detail of the “ring with bezel” cells (hematoxylin–eosin, original magnification: 20×). An example of a ring with bezel cells is indicated with a red arrow.

**Figure 3 diseases-10-00077-f003:**
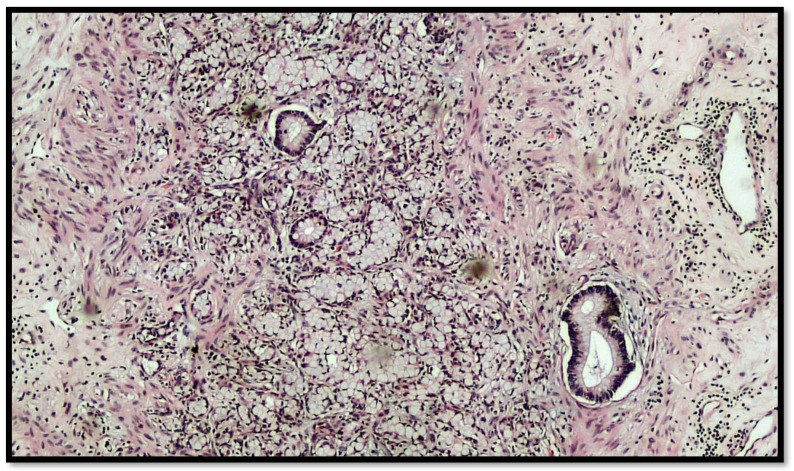
Another photomicrograph showing goblet cell carcinoid and some glands in the muscular wall of the appendix. (Hematoxylin–eosin, original magnification 20×).

**Figure 4 diseases-10-00077-f004:**
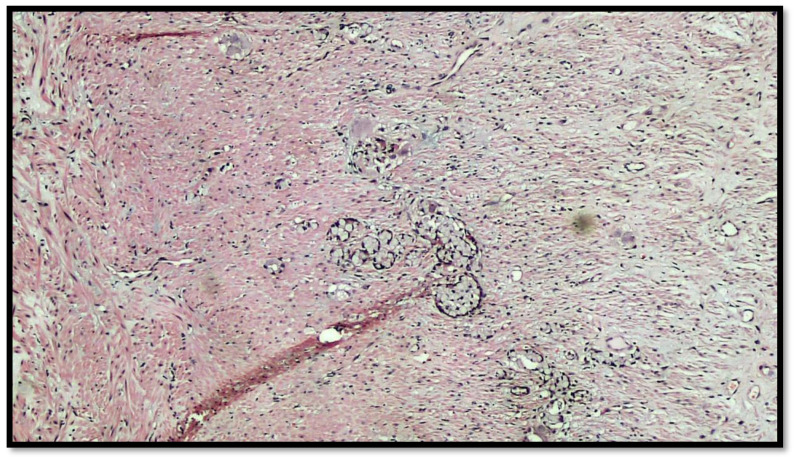
Details of muscular invasion by GCC of the appendix (hematoxylin–eosin, original magnification 20×).

**Figure 5 diseases-10-00077-f005:**
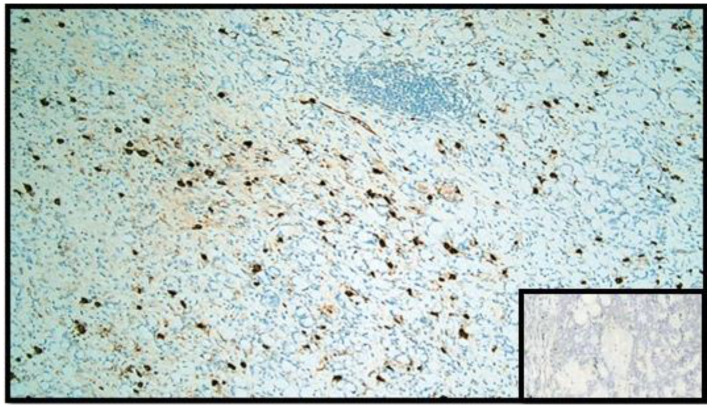
Immunohistochemical reaction with anti-chromogranin A antibody. Note the widespread positivity in the GCC component (IHC, original magnification: 20×). Box: negative control.

**Figure 6 diseases-10-00077-f006:**
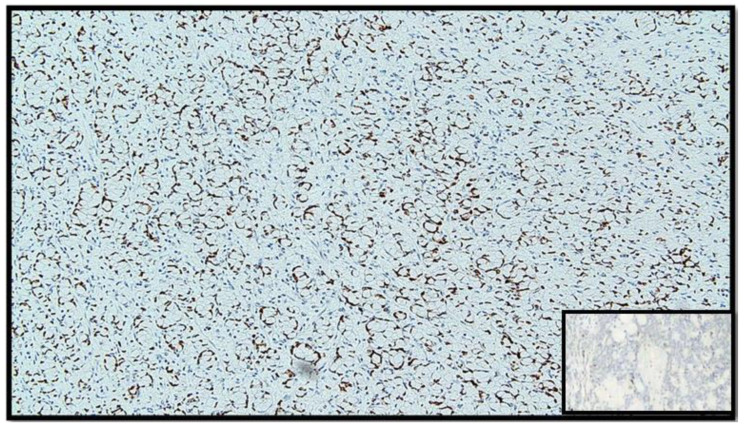
Immunostaining for CDX-2: note the positivity of the nucleus in tumor cells (IHC, original magnification: 40×).

**Table 1 diseases-10-00077-t001:** Summary of clinical history of the patient.

Patient	Time 0	Time 1	Time 2	Time 3
	Ischemic heart disease		Hb: 11 g/dL	Colonoscopy
72 years old woman	Kidney failure	Pleural effusion	Normal CEA	gastroscopy
	Chronic myeloid leukaemia			

## Data Availability

Not applicable.

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
