# Peer review of "Goblet Cell Carcinoma of the Appendix with Synchronous Adenocarcinoma of the Cecum: Distinct or Related Entities?"

_diseases, 2022, doi:10.3390/diseases10040077_

Round 1
Reviewer 1 Report
In this work, the authors, for the first time identified a case of 72yeras old patients having both synchronous colorectal adenocarcinomas of the caecum and Goblet cell carcinoma of the appendix. Through this study, the authors suggest that systematically follow-up is required in order to identify the early onset of second primary malignancies
Critique
This work is interesting, the methodology used is suitable to answer the research question and the discussion is based on sound data. I have only a few questions.
1. Can you please make a table describing the case's medical history?
2. Figure 2. Can you please mark "ring with bezel" cells inside the figure? like, use arrowhead .
Author Response
Dear Reviewer n'1,
first of all thank you very much for your kind congratulations and for your tips.
We have addressed all.
A warm greeting
Reviewer 2 Report
In this research, Leonardo Vincenti et al. discovered Goblet cell carcinoma of the appendix. The authors identified the rare primary tumor of the appendix, which showed the appendix characterized by epithelial and neuroendocrine components. This is the first case of goblet cell carcinoma, and it will be interesting finding for tumor scientists.
The finding is important for cancer researchers and clinicians. However, the manuscript do not provide enough data to illustrate the first finding of GCC of the appendix. The manuscript should be considered for publication, as long as the authors are able to address some specific concerns as follow.
1, the figure 1, HE stain showed the adenocarcinoma arising on villous adenoma. I would like to suggest showing the adenocarcinoma with arrow, and highlighting the typical tumor.
2, Figure 2 showed Globet Cell Carcinoid of the appendix. It is better to show more figures with this Globet Cell Carcinoid and also highlight the carcinoid.
3, figure 3, are there negative tissue picture with anti-Chromogranin A antibody? It is easier to compare with benign sample.
4, I also would like to suggest to show more figures with benign sample as control in figure 4. It is easy to compare and find the tumor characteristic.
5, it is great discovery of Goblet cell carcinoma, so do you have more information such as RNA-seq data, Serum indicators etc, which would be interesting for reader? Why does it difficult to find the GCC of the appendix? In this manuscript, the authors find the first case GCC of the appendix, so how would other scientists to find them for further researchers.
Author Response
Dear Reviewer n'2,
first of all thank you very much for this beautiful compliments. We try to address all your suggestions. Furthermore, we have add some informations about serum markers because, in our case, RNA-seq was not necessary.
A warm greeting,
Round 2
Reviewer 2 Report
The revised manuscript should be accepted. My concerning were solved in the revised manuscript.